# ReferSplat: Referring Segmentation in 3D Gaussian Splatting

**Shuting He** [1]  **Guangquan Jie** [2]  **Changshuo Wang** [3]  **Yun Zhou** [2]  **Shuming Hu** [2]  **Guanbin Li** [4]  **Henghui Ding** [2]

## Abstract

We introduce Referring 3D Gaussian Splatting Segmentation (R3DGS), a new task that aims to segment target objects in a 3D Gaussian scene based on natural language descriptions, which often contain spatial relationships or object attributes. This task requires the model to identify newly described objects that may be occluded or not directly visible in a novel view, posing a significant challenge for 3D multi-modal understanding. Developing this capability is crucial for advancing embodied AI. To support research in this area, we construct the first R3DGS dataset, Ref-LERF. Our analysis reveals that 3D multi-modal understanding and spatial relationship modeling are key challenges for R3DGS. To address these challenges, we propose ReferSplat, a framework that explicitly models 3D Gaussian points with natural language expressions in a spatially aware paradigm. ReferSplat achieves state-of-the-art performance on both the newly proposed R3DGS task and 3D open-vocabulary segmentation benchmarks. Dataset and code are available at https://github.com/heshuting555/ReferSplat.

## 1. Introduction

3D Gaussian Splatting (3DGS) (Kerbl et al., 2023), a recently proposed neural rendering technique, has attracted significant attention because of its fast training, real-time rendering capabilities, and explicit point-based representation (Keetha et al., 2024; Yang et al., 2024; Tang et al., 2024; Zhou et al., 2024a). As 3DGS continues to advance, text-driven 3D scene understanding has attracted more and

---

[1]MoE Key Laboratory of Interdisciplinary Research of Computation and Economics, Shanghai University of Finance and Economics, Shanghai, China [2]Institute of Big Data, College of Computer Science and Artificial Intelligence, Fudan University, Shanghai, China [3]Nanyang Technological University, Singapore [4]Sun Yat-sen University, Guangzhou, China. Correspondence to: Henghui Ding <henghui.ding@gmail.com>.

*Proceedings of the 42nd International Conference on Machine Learning*, Vancouver, Canada. PMLR 267, 2025. Copyright 2025 by the author(s).

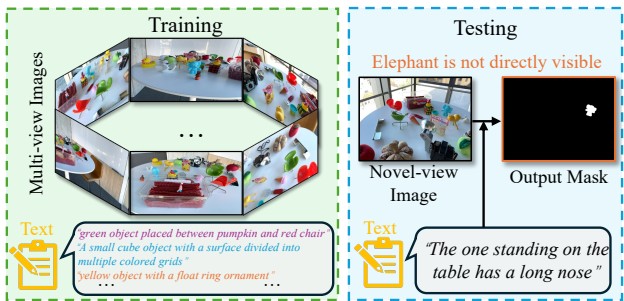

*Figure 1.* Referring 3D Gaussian Splatting Segmentation (R3DGS) aims at segmenting the target objects described by a given natural language descriptions within a 3D Gaussian scene, requiring the model to identify newly described objects that may be occluded or not directly visible in a novel view.

more attention, particularly in open-vocabulary 3DGS segmentation (3DOVS) (Qin et al., 2024; Ye et al., 2025; Zhou et al., 2024b), which relies on fixed-pattern linguistic class names input for segmentation.

However, despite these advancements, free-form natural language interactions with 3D scenes remain underexplored. The ability to interpret and localize objects based on arbitrary language descriptions is crucial for various real-world applications, such as embodied AI (Shorinwa et al., 2024), autonomous driving (Gu et al., 2024), and VR/AR systems (Jiang et al., 2024). To bridge this gap, we introduce a new task: **Referring 3D Gaussian Splatting Segmentation (R3DGS)**, aims at segmenting objects in a 3D Gaussian scene based on natural language expressions that typically encode spatial relationships and descriptive attributes. As shown in Fig. 1, R3DGS requires the model to identify newly described objects, even when occluded or not directly visible in the novel view, posing a significant challenge for 3D multi-modal understanding. To support future research in R3DGS, we construct Ref-LERF, a new dataset rich in complex and spatially grounded language expressions.

A straightforward baseline for R3DGS is to adapt existing open-vocabulary 3D scene understanding methods by replacing short and simple class names with complex natural language expressions. Existing open-vocabulary methods primarily project semantic features onto images for pixel-level understanding, leveraging semantic knowledge from pre-trained 2D vision-language models (Radford et al.,

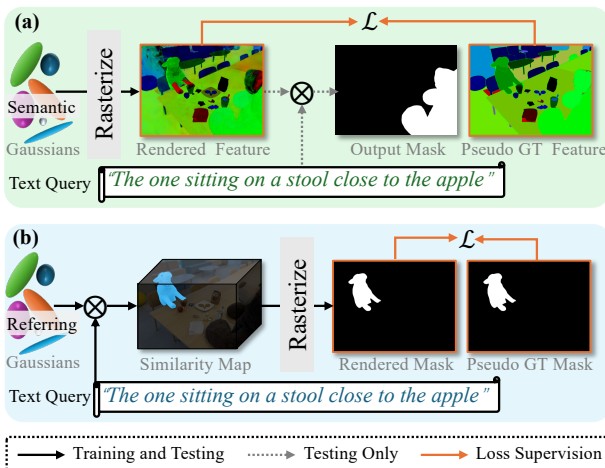

*Figure 2.* Comparison of (a) existing open-vocabulary 3DGS segmentation pipeline and (b) the proposed ReferSplat for R3DGS.

2021; Kirillov et al., 2023) as ground truth feature to guide 3D scene representation learning. During inference, output masks are obtained by matching the input open-vocabulary class names with the rendered feature, as shown in Fig. 2 (a). However, these methods face significant limitations when applied to R3DGS. One major drawback is the lack of interaction between the text query and Gaussian representations during training. Existing methods rely on matching the text query with 2D rendered features instead of performing localization directly in 3D space, which limits their performance in complex scenarios. Moreover, this training paradigm overlooks position information, as rendered feature cannot inherently understand spatial relationships between objects described in a sentence. Instead, it primarily focuses on semantic understanding, making it ineffective for tasks requiring spatial reasoning. This raises a question: *Can we design a network that models 3D Gaussians with language expressions in a spatially aware paradigm?*

In this work, we propose ReferSplat, an end-to-end framework that models 3D Gaussian points with natural language expressions in a spatially aware paradigm for Referring 3D Gaussian Splatting Segmentation (R3DGS). To enable language perception, we assign each 3D Gaussian a referring feature vector, forming referring fields that interact with text queries during training. Similarity is computed between textual features and 3D Gaussian referring features, and the rendered 2D segmentation mask is obtained accordingly, as shown in Fig. 2 (b). For segmentation supervision, we generate pseudo ground truth masks using a confidence-weighted IoU strategy. The constructed 3D Gaussian Referring Fields enable the model to identify objects that are occluded or not directly visible by leveraging 3D scene knowledge learned from multi-view training images. To enhance spatial reasoning, we introduce a Position-aware Cross-Modal Interaction module that extracts position features for both Gaussians and language descriptions. These

features are refined through position-guided attention, effectively aligning 3D Gaussian representations with text descriptions. Despite spatial awareness, sentences with similar semantics but different target objects may cause confusion thus degrade performance. To address this issue, we employ Gaussian-Text Contrastive Learning between positive Gaussian embeddings and text embeddings, where Gaussian embeddings are computed from selectively chosen positive Gaussian referring features. This helps the model differentiate fine-grained referring expressions, improving cross-modal understanding. Extensive experiments demonstrate that ReferSplat achieves state-of-the-art performance on both open-vocabulary 3DGS segmentation and the newly proposed referring 3DGS segmentation tasks.

In summary, our contributions are as follows:

- We introduce a new task termed Referring 3D Gaussian Splatting (R3DGS) and construct a new dataset Ref-LERF to support future research in R3DGS.

- To tackle the challenges of R3DGS, we propose ReferSplat, a spatially aware framework that models 3D Gaussians based on natural language expressions, achieving state-of-the-art performance on Ref-LERF.

- We construct 3D Gaussian Referring Fields, incorporating high-quality pseudo masks generated via confidence-weighted IoU strategy as a strong pipeline.

- We design Position-aware Cross-Modal Interaction module, which integrates position information into the cross-modal interaction, enhancing spatial reasoning and feature alignment between 3D Gaussians and text.

- We employ Gaussian-Text Contrastive Learning to improve the model's ability to generate discriminative multi-modal representations, effectively distinguishing semantically similar expressions.

## 2. Related Work

**3D Neural Representations.** Recent developments in 3D representation have achieved notable progress, with Neural Radiance Fields (NeRF) (Mildenhall et al., 2021) standing out for the ability to generate high-quality novel view synthesis. Despite their effectiveness, NeRF's reliance on implicit neural networks can lead to extended training and rendering durations. To address these limitations, Instant-NGP (Müller et al., 2022) accelerates the process using hash encoding. Recently, 3D Gaussian Splatting (3DGS) (Kerbl et al., 2023) proposes an explicit way to represent 3D scenes using a collection of 3D Gaussian distributions. Through the optimization of their spatial positions and visual attributes, this method achieves real-time, high-quality rendering. Since the introduction of 3DGS, its superior performance has attracted increasing attention, leading to numerous studies focusing on enhancing rendering quality (Li et al., 2025b; Yu et al., 2024a; Huang et al., 2024), improving scene reconstruc-

tion accuracy (Yu et al., 2024b; Dai et al., 2024; Guédon & Lepetit, 2024), and feed-forward optimization (Xu et al., 2025; Chen et al., 2025; Charatan et al., 2024). In this work, we leverage the strengths of 3D Gaussian Splatting as a foundation for 3D neural representations, using its explicit paradigm to achieve efficient and high-quality rendering.

**3D Segmentation in Gaussian Splatting.** Inspired by the success of Gaussian Splatting (Kerbl et al., 2023; Xu et al., 2025; Chen et al., 2025), numerous studies have explored 3D segmentation within this framework. SAGA (Cen et al., 2025) introduces a promptable segmentation method leveraging 3D Gaussian Splatting to highlight its potential in semantic understanding. Following this, various works (Hu et al., 2024; Choi et al., 2025; Jain et al., 2024) emerged, further advancing promptable segmentation techniques. Integrating foundation models such as CLIP (Radford et al., 2021) and SAM (Kirillov et al., 2023; Ravi et al., 2025), several approaches (Li et al., 2025a; Zhou et al., 2024b; Zuo et al., 2024; Ye et al., 2025; Wu et al., 2024b; Liang et al., 2024; Shi et al., 2024; Qu et al., 2024; Ji et al., 2024; Peng et al., 2025) extend Gaussian Splatting to semantic and open-vocabulary 3D segmentation. While these methods incorporate some level of language perception, they primarily focus on category-level segmentation and struggle to comprehend complex natural language queries. This gap underscores the necessity for methods capable of understanding and segmenting objects based on nuanced linguistic cues.

**Referring Segmentation.** 2D Referring Expression Segmentation (RES) (Ding et al., 2023c; 2021; 2023a;b; Liu et al., 2023; Yang et al., 2022; Ding et al., 2025a;b) and 3D point-based Referring Expression Segmentation (3DRES) (Huang et al., 2021; He et al., 2024; 2023; He & Ding, 2024; Wang et al., 2024; 2025) aim to segment a target object within a 2D image or 3D point cloud scene based on a natural language expression. In recent years, RES and 3DRES have seen significant progress, with Transformer-based architectures becoming the dominant choice. Among them, Grounded SAM (Ren et al., 2024) stands out as a robust RES method in the 2D domain, integrating two state-of-the-art models—Grounding DINO (Liu et al., 2025) and SAM (Kirillov et al., 2023). Besides, the emergence of point-based benchmarks such as ScanRefer (Chen et al., 2020) and Multi3DRefer (Zhang et al., 2023) has significantly accelerated the progress of 3DRES. However, despite these advances, existing methods and datasets remain constrained to 2D or 3D point-based representations and cannot be directly applied to R3DGS.

## 3. Method

### 3.1. Preliminaries

In 3D Gaussian Splatting (3DGS) for RGB image rendering, a scene is represented by a set of 3D Gaussian distributions

$\mathcal{G} = \{g_i\}_{i=1}^{\mathcal{N}}$, where $\mathcal{N}$ is the number of Gaussians. Each Gaussian $g_i$ is parameterized by its mean position $\mu_i \in \mathbb{R}^3$, covariance matrix $\Sigma_i \in \mathbb{R}^{3 \times 3}$, opacity $\sigma_i \in \mathbb{R}^1$, and color $c_i \in \mathbb{R}^{d_c}$, where $d_c = 3$ for RGB color parameters.

To render an image, each 3D Gaussian is projected onto the 2D image plane, where it contributes to the pixel color based on its opacity. The color at a given pixel $v$, denoted as $C(v)$, is computed by blending the contributions of all Gaussians according to their opacity. This process is formulated as:

$$C(v) = \sum_{i=1}^{\mathcal{N}} c_i \alpha_i \prod_{j=1}^{i-1} (1 - \alpha_j), \qquad (1)$$

where $c_i$ is the color vector of the $i$-th Gaussian, and $\alpha_i = \sigma_i G_i^{2D}(v)$, $G_i^{2D}(v)$ is the 2D projection function.

### 3.2. Problem Statement and Method Overview

During training for Referring 3D Gaussian Splatting Segmentation (R3DGS), we are given a set of RGB images $\mathcal{I} = \{I_s\}_{s=1}^{S}$ from $S$ different training views, along with a set of natural language expressions $\mathcal{T} = \{T_l\}_{l=1}^{L}$, where each $T_l$ refers to an object visible in a single training view. At test time, the model is given a referring expression and a novel view, and is required to predict the corresponding object mask. The key challenge lies in segmenting the target object in this unseen view, where it may be partially occluded or even entirely invisible. Unlike 2D referring segmentation, which operates on a single image and cannot resolve occlusions, R3DGS leverages the 3D scene learned from multi-view supervision to reason about occluded objects based on their visibility in other training views. Compared to 3D referring segmentation, which relies on explicit point clouds and annotated 3D masks, R3DGS aims to learn from multi-view 2D training images without requiring annotated mask supervision. This setting makes the proposed R3DGS task more practical for real-world applications.

An overview of our proposed approach ReferSplat is shown in Fig. 3. The approach begins with the construction of 3D Gaussian Referring Fields. In Sec. 3.3, we extract word features $f_w$ and a sentence embedding $f_e$ from the referring expression $T_l$ using a text encoder. To infuse language-awareness into the 3D Gaussians, we introduce a new property called referring features. The segmentation mask is then obtained by modulating these referring features with word features $f_w$, followed by rendering and supervision with the pseudo-mask generated. To enhance the interaction between referring features and word features $f_w$, we introduce a Position-aware Cross-Modal Interaction in Sec. 3.4, which strengthens the alignment between spatial and linguistic cues. Finally, in Sec. 3.5, we employ contrastive learning to distinguish between different text queries by enforcing semantic consistency between positive Gaussian embeddings with $f_e$, leading to a more robust language-guided

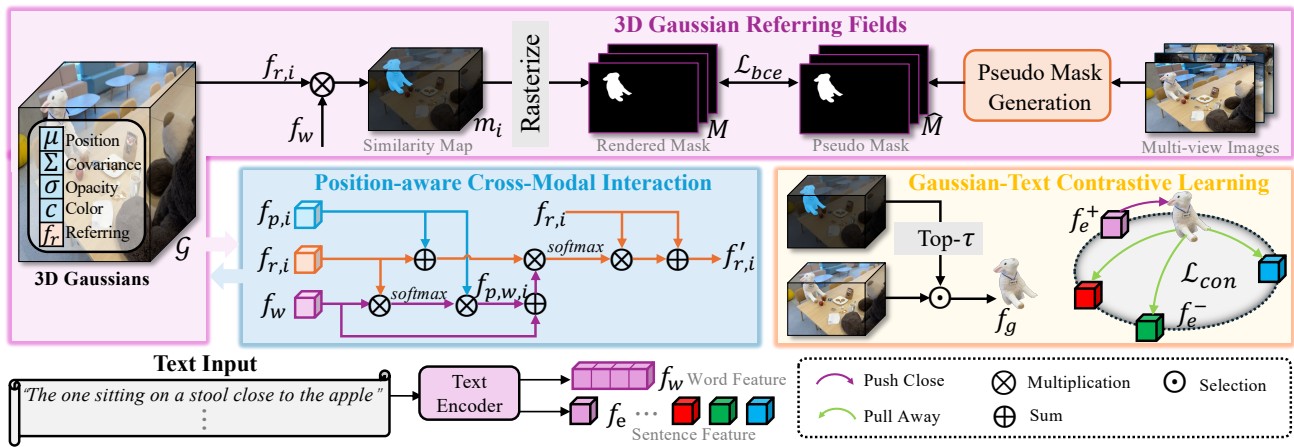

**Figure 3.** Overview of the proposed approach **ReferSplat**. Firstly, to infuse language-awareness into the 3D Gaussians, we introduce a new property called referring features, constructing 3D Gaussian Referring Fields. The segmentation mask is then obtained by modulating these referring features with word features $f_w$, followed by rendering and supervision with the generated pseudo-mask. To enhance the interaction between referring features and word features $f_w$, we introduce a Position-aware Cross-Modal Interaction. Finally, we employ contrastive learning to distinguish semantically similar language expressions.

segmentation in 3D Gaussian scene.

### 3.3. 3D Gaussian Referring Fields

Inspired by methods that incorporate semantic feature vectors to construct semantic-aware fields (Qin et al., 2024; Zhou et al., 2024b; Qu et al., 2024), we introduce a referring feature vector $f_{r,i} \in \mathbb{R}^{d_r}$ for each 3D Gaussian, where $d_r$ denotes the feature dimension. This forms referring fields that equip 3D Gaussians with language perception capabilities. In our framework, the referring feature encodes semantic and referring information, allowing us to compute the text response for each Gaussian by measuring similarity between the 3D Gaussian referring feature and the input text. Unlike LangSplat (Qin et al., 2024), which performs retrieval-based matching between rendered semantic features and text embeddings, we directly model the relationship between 3D Gaussians and text embeddings. Specifically, we compute the similarity between each Gaussian referring feature and the input sentence representation by aggregating responses across all words:

$$m_i = \sum_j f_{r,i} \times f_{w,j}, \qquad (2)$$

where $m_i \in \mathbb{R}^1$ represents the overall Gaussian-language similarity score for the $i$-th Gaussian, and $f_{w,j}$ denotes the feature representation of the $j$-th word in the sentence. To ensure computational efficiency, both features are mapped to the same feature dimension, denoted as $D$.

Next, similar to traditional Gaussian Splatting, we apply a rasterization process. Instead of rendering RGB values, we rasterize $m_i$. The response on the 2D image plane at pixel $v$, denoted as $M(v)$, serves as both the projected text response

and the rendered segmentation mask, computed as:

$$M(v) = \sum_{i=1}^{\mathcal{N}} m_i \alpha_i \prod_{j=1}^{i-1} (1 - \alpha_j). \qquad (3)$$

Finally, we employ a binary cross-entropy (BCE) loss to supervise the output mask, enforcing consistency with the pseudo ground truth mask, which we introduce later:

$$\mathcal{L}_{bce} = -\sum_v \left[ \hat{y} \log y + (1 - \hat{y}) \log(1 - y) \right], \qquad (4)$$

where $\hat{y}$ represents the pseudo ground truth mask $\hat{M}(v)$, and $y$ denotes the predicted mask $M(v)$ at pixel $v$.

Unlike traditional 3DGS, which primarily renders color values or predefined semantic features, our approach directly renders Gaussian-language similarity responses, enabling explicit interaction between textual descriptions and 3D scene representations. As shown in Tab. 2, our method surpasses existing approaches, establishing a superior referring segmentation framework in 3D Gaussian scenes. Additionally, our constructed 3D Gaussian Referring Fields empower the model to recognize occluded or non-visible objects by leveraging 3D scene knowledge learned from multi-view training images.

**Pseudo Mask Generation.** To generate high-quality 2D pseudo masks, we input the image and referring expression into Grounded SAM (Ren et al., 2024). Since a single object can be described by multiple language expressions, this process produces $K$ candidate masks $\{\hat{M}_k\}_{k=1}^K$, each assigned a confidence score $\gamma$ that exceeds a predefined threshold $\epsilon$. A naive approach selects the highest-confidence mask, but this often fails as confidence does not always correlate

with correctness. Our analysis reveals that while the correct mask is usually present, it does not always receive the highest confidence score.

To improve selection, we first compute the Intersection over Union (IoU) consistency across all $K$ candidate masks and select the most frequently occurring or overlapping mask. However, this purely IoU-based approach disregards confidence scores, potentially favoring low-quality masks. To address this, we propose a confidence-weighted IoU strategy, defining an overlapping score $p_k$ as:

$$p_k = \sum_{j=1}^{K} \frac{\left(\gamma_k \hat{M}_k \cap \gamma_j \hat{M}_j\right)}{\left(\gamma_k \hat{M}_k \cup \gamma_j \hat{M}_j\right)}, \qquad (5)$$

where $\hat{M}_k$ and $\hat{M}_j$ are candidate masks with confidence scores $\gamma_k$ and $\gamma_j$, respectively. This formulation prioritizes masks with both high IoU consistency and confidence. Finally, we select the mask with the highest $p_k$ as the final pseudo-mask $\hat{M}$. Our pseudo Mask Generation strategy significantly improves mask quality, enhancing accuracy and robustness in referring segmentation in 3D Gaussians.

### 3.4. Position-aware Cross-Modal Interaction

The accuracy of the output mask is determined by the similarity score $m_i$ in Eq.2 that measures the relationship between Gaussian referring features and text embeddings. To improve the accuracy of similarity score, it is essential to establish explicit cross-modal interactions, facilitating information exchange and effectively leveraging cross-modal dependencies for enhanced semantic understanding. Furthermore, Gaussian referring features primarily capture semantic information, which helps recognize object categories and attributes. However, comprehending a sentence like *"the red object on the left"* requires not only semantic recognition (*"red object"*) but also spatial reasoning (*"on the left"*). Therefore, position information, which represents spatial locations and object relationships, is crucial for accurately understanding language expressions and achieving precise object segmentation within 3D Gaussian representations.

To address these issues, we propose a Position-aware Cross-Modal Interaction module that injects position information into the cross-modal attention mechanism to facilitate interactions between textual entities and 3D Gaussians beyond mere semantic alignment. This module enables a richer and mutually informed fusion of spatial and semantic information, strengthening the alignment between 3D Gaussians and text embeddings. In this framework, position, semantics, and language are interdependent: position provides spatial context for semantics, while semantic information aids in understanding spatial relationships, ensuring a coherent and context-aware cross-modal representation.

**Position Feature Extraction.** To integrate position infor-

mation, we first extract position features from 3D Gaussian representations. Specifically, the center coordinates $\{\mu_i\}_{i=1}^{\mathcal{N}}$ of Gaussians are projected through an MLP layer to obtain position embeddings $f_{p,i} \in \mathbb{R}^D$, serving as position-aware representations of Gaussian referring features. Meanwhile, understanding spatial relationships from language descriptions is important. However, unlike 3D Gaussians, text features lack explicit positional information. Since language expressions often describe objects in relation to one another, we infer text position information from the position embeddings of Gaussians by leveraging their cross-modal correspondence. To achieve this, we first compute the relationship between word features $f_w$ and Gaussian referring features $f_{r,i}$. Using this relationship, we extract the corresponding position information from Gaussian position features $f_{p,i}$, formulated as:

$$f_{p,w,i} = \text{softmax}\left(\frac{f_w f_{r,i}^T}{\sqrt{D}}\right) f_{p,i}, \qquad (6)$$

where $D$ is the feature dimension, and $f_{p,w,i}$ represents the position feature aligned with the word features. In this way, the model dynamically extracts text position information based on textual descriptions and Gaussian referring cues.

**Position-aware Attention for Feature Refinement**. After obtaining position information $f_{p,i}$ for Gaussian referring features and $f_{p,w,i}$ for text embeddings, we further refine the referring feature representation using a position-guided attention mechanism. We integrate structural geometry constraints to guide attention computation:

$$f'_{r,i} = f_{r,i} + \text{softmax}\left(\frac{(f_{r,i} + f_{p,i})(f_w + f_{p,w,i})^T}{\sqrt{D}}\right) f_w. \qquad (7)$$

This formulation ensures that the updated referring feature $f'_{r,i}$ is enriched with both position and semantic cues, improving its ability to localize the target object. Besides, we retain the original $f_{r,i}$ to preserve fine-grained details that may be lost in the attention process. Then, $f'_{r,i}$ replace $f_{r,i}$ in Eq.2 to obtain more accurate text response. The proposed Position-aware Cross-Modal Interaction module establishes a stronger relationship between Gaussian referring features and text descriptions by explicitly integrating position information. This integration enhances spatial understanding, and strengthens cross-modal feature alignment, enabling more precise R3DGS.

### 3.5. Gaussian-Text Contrastive Learning

While the proposed Position-aware Cross-Modal Interaction module effectively captures the relationship between Gaussian representations and text descriptions, distinguishing between languages with similar meanings but referring to different objects remains a significant challenge. Such

ambiguities can lead to confusion in referring segmentation, particularly in complex 3D Gaussian scenes. To address this issue, we introduce Gaussian-Text Contrastive Learning in the Gaussian feature space. This learning paradigm enhances the model's ability to disambiguate similar language expressions that refer to different objects, enforcing a stronger Gaussian-language understanding and alignment for more robust referring segmentation.

A key challenge of Gaussian-Text Contrastive Learning lies in extracting positive Gaussian referring features that accurately match the text description. In a typical 3D Gaussian scene, numerous Gaussians exist, and 3D mask annotations for precisely outlining the Gaussians corresponding to the text are unavailable. However, from Eq.2, we can obtain the response $m_i$ of the text on the 3D Gaussians and select Gaussian points with high similarity. Therefore, we consider Gaussian referring features whose $m_i$ are within the top-$\tau$ percentile as positive examples. The positive Gaussian embedding, denoted as $f_g$, corresponding to the text is then obtained by averaging the features of these chosen Gaussian referring features:

$$f_g = \frac{1}{N_\tau} \sum_{i \in \mathbb{N}_\tau} f'_{r,i}, \qquad (8)$$

where $\mathbb{N}_\tau$ denotes the set of indices corresponding to the top-$\tau$ percentile of $m_i$, and $N_\tau$ is its cardinality.

After obtaining paired Gaussian embedding and text embedding, we employ object-wise contrastive learning:

$$\mathcal{L}_{con} = -\frac{1}{|\mathbf{P}|} \sum_{f_e^+ \in \mathbf{P}} \log \frac{\exp(f_g \cdot f_e^+)}{\sum_{f'_e \in \mathbf{P},\mathbf{N}} \exp(f_g \cdot f'_e)}, \quad (9)$$

where $\mathbf{P}$ is the set for positive textual samples with positive Gaussian embedding and $\mathbf{N}$ represents the set of negative textual samples, which are drawn from different text queries in the scene. This formulation encourages the model to maximize similarity between Gaussians and their corresponding textual descriptions while ensuring sufficient separation from unrelated textual descriptions, ultimately enhancing accurate referring segmentation in 3D Gaussian Splatting. The total training objective is:

$$\mathcal{L}_{loss} = \mathcal{L}_{bce} + \lambda \mathcal{L}_{con}, \qquad (10)$$

where $\lambda$ is used for balancing the contrastive loss $\mathcal{L}_{con}$.

Following (Mirzaei et al., 2023), we adopt a two-stage optimization approach to further refine segmentation masks. Instead of using the pseudo masks from Sec. 3.3, we leverage the rendered masks generated by our trained RefraSplat model in the first stage to supervise a secondary ReferSplat model. This progressive refinement provides more accurate initial masks, improving segmentation performance.

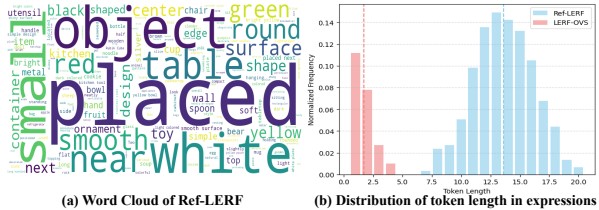

(a) Word Cloud of Ref-LERF    (b) Distribution of token length in expressions

*Figure 4.* Dataset analysis of our constructed Ref-LERF.

## 4. Experiments

### 4.1. Ref-LERF Dataset and Evaluation Metrics

The LERF dataset (Kerr et al., 2023) is collected using the Polycam iPhone app and consists of four diverse, complex, real-world scenes. It was originally developed for 3D object localization tasks. LangSplat (Qin et al., 2024) introduces ground truth (GT) mask annotations to enhance its complexity, forming the LERF-OVS dataset for 3D open-vocabulary segmentation. We introduce Ref-LERF, incorporating language expression annotations to further expand its capabilities to enable R3DGS evaluation. Each scene contains approximately five expressions per object, with 236 language descriptions used for training and 59 for testing, totaling 295 descriptions for 59 objects. Besides, annotations emphasize positional relationships, providing richer contextual grounding for precise object segmentation.

**Dataset Analysis.** The word cloud of our newly proposed Ref-LERF dataset, visualized in Fig. 4 (a), highlights its richness in spatial and detailed descriptions. A significant portion of the dataset consists of relative position words such as "placed", "near", and "next", as well as fine-grained object attributes like "round" and "surface". This demonstrates that Ref-LERF places a stronger emphasis on spatial reasoning and detailed object understanding compared to previous datasets. Furthermore, as illustrated in Fig. 4 (b), Ref-LERF presents a significantly greater challenge than LERF-OVS, featuring more complex sentences and richer object descriptions. The average sentence length exceeds 13.6 words, making it approximately eight times longer than those in LERF-OVS. This increased complexity ensures that models trained on Ref-LERF must develop a deeper understanding of language and spatial relationships, making it a more realistic and comprehensive benchmark for R3DGS.

**Evaluation Metrics.** The average IoU (mIoU) is calculated between the masks rendered from the text response on the 3D Gaussians and the GT object masks.

### 4.2. Implementation Details

We extract text embeddings for each sentence using BERT (Devlin et al., 2019). Following the default configuration of LangSplat (Qin et al., 2024), we first train the RGB

*Table 1.* Ablation study on our method. PCMI, and GTCL denote components of Position-aware Cross-Modal Interaction, and Gaussian-Text Contrastive Learning, respectively.

| | Components | | Results | |
| --- | --- | --- | --- | --- |
| Index | PCMI | GTCL | ramen | kitchen |
| Baseline | ✗ | ✗ | 28.4 | 18.5 |
| 1 | ✓ | ✗ | 33.5 | 22.8 |
| 2 | ✗ | ✓ | 32.8 | 21.9 |
| Ours | ✓ | ✓ | 35.2 | 24.4 |
| Two-stage | ✓ | ✓ | 36.9 | 25.2 |

*Table 2.* Ablation study on Baseline Configuration.

| | Results | |
| --- | --- | --- |
| Method | ramen | kitchen |
| LangSplat | 12.0 | 17.9 |
| SPIn-NeRF | 7.3 | 10.3 |
| Cosine Similarity | 25.4 | 16.8 |
| Multiplication (Ours) | 28.4 | 18.5 |

*Table 3.* Ablation study on Pseudo Mask Generation.

| | Results | | Mask Quality | |
| --- | --- | --- | --- | --- |
| Method | ramen | kitchen | ramen | kitchen |
| Top-1 | 23.1 | 15.7 | 47.1 | 42.8 |
| SAM2 | 22.6 | 15.5 | 41.2 | 37.8 |
| IoU w/o $\gamma$ | 25.3 | 17.1 | 48.2 | 45.3 |
| IoU w/ $\gamma$ (Ours) | 28.4 | 18.5 | 52.9 | 49.7 |

*Table 4.* Ablation study on Cross-Modal Interaction Design.

| | Results | |
| --- | --- | --- |
| Method | ramen | kitchen |
| w/o $f_{p,i}$ and $f_{p,w,i}$ | 27.9 | 18.1 |
| w/o $f_{p,w,i}$ | 29.2 | 18.8 |
| $f_{p,i} + f_{r,i}$ | 30.3 | 20.5 |
| w/ $f_{p,i}$ and $f_{p,w,i}$ (Ours) | 33.5 | 22.8 |

representation of the 3D scene and then freeze its parameters before training the components of our proposed model. We optimize the Gaussian referring features for 45,000 iterations, using a learning rate of 0.0025, while other parameters, such as the MLP, are trained with a learning rate of 0.0001. The Adam optimizer (Kingma & Ba, 2015) is used for optimization. To enable gradient back-propagation through extended referring feature attributes, we modify the CUDA kernel to render referring features on 3D Gaussians. Training is conducted on an NVIDIA RTX A6000 GPU. For hyper-parameter optimization, we set $d_r$, $D$, $\epsilon$, and $\lambda$ to 16, 128, 0.3, and 0.02, respectively. While $\tau$ varies with training, following the schedule: $\tau = 0.1 \times 0.6^{(iteration/1000)}$.

### 4.3. Ablation Study

The following ablation studies are conducted on the ramen and kitchen scenes on the Ref-LERF dataset.

**Module Effectiveness**. We conduct ablation experiments to evaluate the effectiveness of different components. As shown in Tab. 1, incorporating PCMI (index 1) improves mIoU by 5.1% and 4.3%, respectively compared to the baseline, which is our constructed Referring Feature Fields. This demonstrates that PCMI enhances spatial reasoning and strengthens feature alignment between 3D Gaussians and text. Next, we introduce Gaussian-Text Contrastive Learning (GTCL, index 2) to construct discriminative multimodal representations, improving the model's ability to distinguish semantically similar expressions. Incorporating GTCL leads to 4.4% and 3.4% mIoU improvement, respectively. When integrating all components (index 3), referred to as ReferSplat, we achieve a substantial performance gain, reaching a new state-of-the-art. This result demonstrates the effectiveness of our proposed approach. Furthermore,

our two-stage optimization in Sec. 3.5 further refines the generated masks through ReferSplat trained in the previous stage, improving 3D scene understanding and ensuring more consistent and reliable segmentation results.

**Baseline Evaluation**. We set several baselines for comparison: 1) LangSplat Adaptation: we modify LangSplat (Qin et al., 2024) from open-vocabulary segmentation to referring segmentation by replacing phrase inputs with referring expressions during testing. 2) SPIn-NeRF Adaptation: we adapt SPIn-NeRF (Mirzaei et al., 2023) from prompt-based segmentation to referring segmentation by incorporating text input and summing textual features with original semantic features to generate masks. 3) Cosine Similarity: we compute similarity $m_i$ in Eq.2 using cosine similarity for text response. The results in Tab. 2 show that our method significantly outperforms all baselines, demonstrating that the proposed 3D Referring Feature Fields effectively models the relationship between 3D Gaussians and text.

**Pseudo Mask Generation.** To assess the quality of the generated pseudo labels and support further research, we manually annotate ground truth masks for comparison. As shown in Tab. 3, our method achieves about 50% mIoU against ground truth, validating the effectiveness of our pseudo mask generation. In contrast, alternative approaches—such as using the top-1 prediction, propagating the first-frame mask with SAM2 (Ravi et al., 2025), or selecting masks solely based on IoU without confidence weighting—yield inferior results. These findings underscore our superiority.

**Position-aware Cross-Modal Interaction Design.** We conduct experiments to evaluate the impact of different cross-modal interaction designs based on baseline. As shown in Tab. 4, removing components $f_{p,i}$ and $f_{p,w,i}$ from Eq.7 results in performance dropping below the baseline, indicating that vanilla cross-attention alone is ineffective for our task. Adding $f_{p,i}$ without $f_{p,w,i}$ still yields unsatisfactory

*Table 5.* Analysis of Computation Costs.

| Method | Training | FPS | Storage | mIoU |
|---|---|---|---|---|
| LangSplat | 176min | 12.4 | 46MB | 13.9 |
| GS-Grouping | 66min | **54.2** | **2.3MB** | 14.4 |
| **ReferSplat** (Ours) | **58min** | 26.8 | 3.3MB | **29.2** |

*Table 6.* Choice of Language Encoder.

| Method | ram. | fig. | tea. | kit. | avg. |
|---|---|---|---|---|---|
| CLIP | 23.5 | 23.2 | 26.2 | 21.0 | 23.5 |
| BERT | 35.2 | 25.7 | 31.3 | 24.4 | 29.2 |

*Table 7.* Ablation study on number of feature dims.

| Number | Results | |
|---|---|---|
| | ramen | kitchen |
| 1 | 21.7 | 13.2 |
| 4 | 23.1 | 15.4 |
| 16 | 28.4 | 18.5 |
| 32 | 27.2 | 16.9 |

*Table 8.* R3DGS result on the Ref-LERF dataset.

| Method | ram. | fig. | tea. | kit. | avg. |
|---|---|---|---|---|---|
| Grounded SAM | 14.1 | 16.0 | 16.9 | 16.2 | 15.8 |
| LangSplat | 12.0 | 17.9 | 7.6 | 17.9 | 13.9 |
| SPIn-NeRF | 7.3 | 9.7 | 11.7 | 10.3 | 9.8 |
| GS-Grouping | 27.9 | 8.6 | 14.8 | 6.3 | 14.4 |
| GOI | 27.1 | 16.5 | 22.9 | 15.7 | 20.5 |
| **ReferSplat** (Ours) | **35.2** | **25.7** | **31.3** | **24.4** | **29.2** |

results, suggesting that both components contribute to effective interaction. Additionally, we analyze the impact of directly incorporating position information $f_{p,i}$ into the $f_{r,i}$. Although this improves performance, it remains less effective than our attention-based position modeling. Therefore, our proposed method is crucial for accurate R3DGS.

**Analysis of Computation Costs.** We conduct experiments on the ramen scene from the Ref-LERF dataset using the same NVIDIA A6000 GPU to compare the computational cost of our ReferSplat against SOTA methods in Tab. 5. Results show that ReferSplat achieves significantly lower computational complexity and faster inference speed than LangSplat (Qin et al., 2024). While GS-Grouping (Ye et al., 2025) excels in storage and FPS, ReferSplat outperforms all methods in segmentation performance. ReferSplat also has the shortest training time, thanks to a lightweight preprocessing pipeline that avoids costly operations like language feature compression (LangSplat) or mask association with video tracking methods (GS-Grouping). These results demonstrate that ReferSplat's compact, efficient design is well-suited for real-world applications.

**Choice of Language Encoder.** we conduct experiments comparing BERT and CLIP embeddings for language features in R3DGS in Tab. 6. Results show that BERT consistently outperforms CLIP. This is likely because CLIP focuses more on noun categories, while referring expressions often involve spatial and attribute-based descriptions.

**Impact of Referring Feature Dimension.** We study the effect of the referring feature dimension $d_r$ in Tab. 7 for each 3D Gaussian. In our experiments, we set $d_r$ to 1, 4, 16, and 32, and find that 16 achieves the best results. Smaller dimensions (e.g., 1 or 4) lack the capacity to store discriminative features, while larger dimensions (e.g., 32) introduce redundancy and noise, degrading performance.

### 4.4. Results on the Ref-LERF Dataset

**Quantitative Results**. We evaluate ReferSplat against state-of-the-art 2D and 3D methods on the Ref-LERF dataset in Tab. 8. ReferSplat outperforms 2D-based methods like

Grounded SAM (Ren et al., 2024) and 3D-based approaches such as LangSplat (Qin et al., 2024) and Gaussian Grouping (Ye et al., 2025), demonstrating its effectiveness. Our 3D Gaussian Referring Fields enable the model to recognize occluded or non-visible objects by leveraging multi-view 3D scene knowledge—an inherent limitation of 2D-based methods. While Grounded-SAM generates high-quality masks during training, it is restricted to visible objects within a single view (Sec. 3.2). Additionally, our position-aware cross-modal interaction and contrastive learning enhance spatial reasoning and feature alignment, improving language comprehension in complex 3D environments.

**Qualitative Visualization**. Fig. 5 qualitatively presents that ReferSplat effectively captures the spatial relationship between Gaussian points and text, enabling superior segmentation even in challenging scenarios with heavy occlusion or non-visible objects, as illustrated in (a) and (b).

### 4.5. 3D Open-Vocabulary Segmentation Result

To further validate our method, we evaluate it on 3D open-vocabulary segmentation benchmarks, as shown in Tab. 9 and 10. Despite not being specifically designed for 3DOVS tasks, our approach achieves state-of-the-art performance. This can be attributed to the integration of 3D Gaussian Referring Fields, position-aware cross-modal interaction, and gaussian-text contrastive learning, which enhance spatial reasoning and feature alignment, significantly improving language comprehension in complex 3D environments.

## 5. Conclusion

We introduce Referring 3D Gaussian Splatting Segmentation (R3DGS), a new task for segmenting target objects in 3D Gaussian scenes using natural language descriptions involving spatial relations and object properties. To support research in this area, we construct the first R3DGS

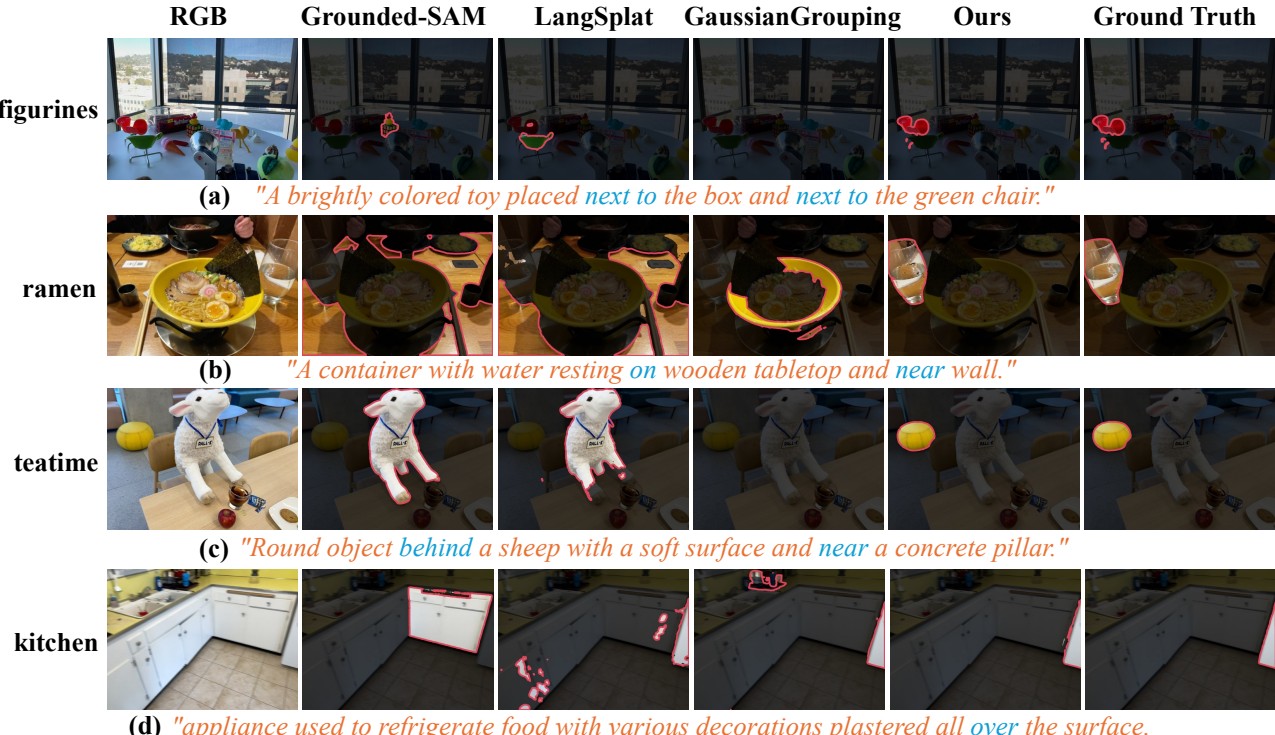

*Figure 5.* Qualitative R3DGS comparisons on the Ref-LERF dataset. Blue masks represent spatial descriptions.

*Table 9.* Open-vocabulary segmentation result on the LERF-OVS.

| Method | ram. | fig. | tea. | kit. | avg. |
|---|---|---|---|---|---|
| Feature-3DGS | 43.7 | 58.8 | 40.5 | 39.6 | 45.7 |
| LEGaussians | 46.0 | 60.3 | 40.8 | 39.4 | 46.6 |
| LangSplat | 51.2 | 65.1 | 44.7 | 44.5 | 51.4 |
| GS-Grouping | 45.5 | 60.9 | 40.0 | 38.7 | 46.3 |
| GOI | 52.6 | 63.7 | 44.5 | 41.4 | 50.6 |
| **ReferSplat** (Ours) | **55.1** | **67.5** | **50.1** | **48.9** | **55.4** |

*Table 10.* Open-vocabulary segmentation result on the 3D-OVS.

| Method | bed | bench | room | sofa | lawn | avg. |
|---|---|---|---|---|---|---|
| Feature-3DGS | 83.5 | 90.7 | 84.7 | 86.9 | 93.4 | 87.8 |
| LEGaussians | 84.9 | 91.1 | 86.0 | 87.8 | 92.5 | 88.5 |
| LangSplat | 92.5 | 94.2 | 94.1 | 90.0 | 96.1 | 93.4 |
| GS-Grouping | 83.0 | 91.5 | 85.9 | 87.3 | 90.6 | 87.7 |
| GOI | 89.4 | 92.8 | 91.3 | 85.6 | 94.1 | 90.6 |
| **ReferSplat** (Ours) | **93.2** | **94.8** | **94.6** | **91.8** | **96.5** | **94.1** |

dataset Ref-LERF. To address R3DGS challenges, we propose ReferSplat which spatially models 3D Gaussian points based on natural language expression. Experiments show that the proposed ReferSplat achieves state-of-the-art performance on both R3DGS and 3DOVS tasks.

## 6. Limitation and Future Work

1) Our current method does not account for dynamic factors, which are crucial for real-world applications. Integrating our approach with 4D Gaussian Splatting (4DGS) (Wu et al., 2024a) could enhance its capability to handle temporal variations and dynamic environments. 2) While we focus on 3D referring segmentation in Gaussian Splatting, our method does not incorporate 3D visual grounding. Extending our framework to support precise object size estimation could further improve its applicability in spatial reasoning and real-world localization tasks. 3) The current dataset includes a

limited number of scenes, which restricts the model's ability to achieve the same robust generalization as 2D-based approaches. In the future, we aim to construct a large-scale dataset, enabling better scene diversity and representation learning, thereby advancing research in this field.

## Impact Statement

This paper presents work whose goal is to advance the field of Machine Learning. There are many potential societal consequences of our work, none which we feel must be specifically highlighted here.

## Acknowledgements

This project was supported by the National Natural Science Foundation of China (NSFC) under Grant No. 62472104, Shanghai Pujiang Programme 24PJD030, Natural Science Foundation of Shanghai 25ZR1402138, and partially supported by NSFC Grant No. 62322608.

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
