# OpenReview forum: "ReferSplat: Referring Segmentation in 3D Gaussian Splatting"
_ICML.cc/2025/Conference — ICML 2025 oral_

### Official Review · Reviewer_naU6 · 2025-02-18

**Overall Recommendation:** 4

**Summary:**

This paper introduces a new task—Referring 3D Gaussian Splatting Segmentation (R3DGS), which aims to segment target objects in 3D Gaussian Splatting scenes based on natural language descriptions. The authors construct the dataset specifically for this task, named Ref-LERF, and propose a framework called ReferSplat. The method primarily builds 3D Gaussian Referring Fields, introduces a Position-aware Cross-Modal Interaction module to fuse spatial and textual information, and employs Gaussian-Text Contrastive Learning to enhance the discriminative capability of cross-modal features, achieving state-of-the-art performance on both the R3DGS and 3D open-vocabulary segmentation tasks.

**Claims And Evidence:**

The authors claim that ReferSplat achieves SOTA performance through extensive comparisons with existing methods. The effectiveness of individual modules (e.g., PCMI and GTCL) is demonstrated.

**Essential References Not Discussed:**

No.

**Experimental Designs Or Analyses:**

The authors perform extensive comparisons with existing methods (such as LangSplat and Grounded SAM), providing both quantitative and qualitative analyses.
However,

**Methods And Evaluation Criteria:**

Methodologically, the authors cleverly incorporate natural language guidance into the conventional 3D Gaussian Splatting framework, enabling the model to capture fine-grained segmentation details of target objects in 3D scenes. The evaluation metrics, such as mean IoU, are standard, and the constructed Ref-LERF dataset provides a reasonable test platform for the task.

**Other Comments Or Suggestions:**

No.

**Other Strengths And Weaknesses:**

### Strengths
1. Provides comprehensive ablation studies and quantitative comparisons, with experimental results showing significant performance improvements.

### Weaknesses
1. The model is trained separately for each scene, which might affect its scalability and generalization ability in large-scale deployments;
2. The dataset is relatively small, and more scenes will be needed in the future to validate the robustness of the method;
3. Refer to "Relation To Broader Scientific Literature*"

**Questions For Authors:**

Why does incorporating positional information into the refer feature (CLIP feature) in Eq. (7) not degrade open-vocabulary segmentation performance, but instead improve it?

**Relation To Broader Scientific Literature:**

This work is closely related to the latest developments in 3D neural rendering.
However, for the R3DGS task, GOI[1] stands out from other methods by incorporating a 2D referring expression segmentation model to assist in 3D localization, making it one of the most comparable works to your proposed approach. However, the current submission does not include a  comparison with GOI in Ref-LeRF dataset. I recommend conducting and reporting such a comparison—or at least providing a detailed discussion regarding how your method differs from GOI—to strengthen the experimental validation and highlight the relative advantages of your proposed framework.

[1] Goi: Find 3d gaussians of interest with an optimizable open-vocabulary semantic-space hyperplane.

**Theoretical Claims:**

The paper presents several derivations regarding rendering formulas, cross-modal attention mechanisms, and contrastive learning. The overall theoretical derivations are consistent with commonly used methods in the field.

---

> ### Author Rebuttal · Authors · 2025-04-01
>
> We sincerely appreciate your positive feedback on our work: clever, reasonable, effective PCMI and GTCL design, and comprehensive ablation studies.
>
> >**Q1&Q2: Generalization ability**
>
> **A1:** The experiments in our main paper follow a per-scene optimization setup, which naturally limits direct generalization to unseen scenes. To address this, we explore a generalized training paradigm, as shown in **Tab. 11 of the Appendix**. In this setting, the referring feature is no longer per-scene initialized; instead, it is predicted from other Gaussian attributes such as color, opacity, and position, following a feed-forward paradigm. This design enables generalization beyond per-scene optimization. To further validate the generalization capability of our method, we conduct additional experiments on the larger-scale and more diverse ScanNet dataset. We select 30 scenes from the official training split for joint training and 5 scenes from the validation split for evaluation. Language expressions are sourced from ScanRefer. As shown in the table below, our method achieves strong performance across diverse and unseen environments, highlighting its robustness and generalization. We expect that extending our framework with scalable architectures (e.g., MVSplat [a]) will lead to further performance gains and improved applicability in real-world scenarios.
> |Method|scene0011|scene0015|scene0019| scene0025|scene0030|mIoU|
> |-|-|-|-|-|-|-|
> |ReferSplat|15.9|19.7|27.8|18.3|21.4|20.6
>
> [a] MVSplat: Efficient 3D Gaussian Splatting from Sparse Multi-View Images, ECCV 2024.
> >**Q3: GOI result**
>
> **A3:** Thank you for this insightful suggestion. To further validate the effectiveness of our method, we have conducted additional experiments comparing our approach with GOI [b] on the Ref-LERF dataset. The results demonstrate that although GOI outperforms LangSplat, its performance remains notably lower than that of ReferSplat. This highlights that, although both methods utilizing 2D pseudo-mask supervision, the key difference lies in how effectively the connection between 3D Gaussian points and natural language expressions is established within the 3D scene. We will revise the main paper to discuss GOI [b] and update Tab. 5 with the new comparison results shown below.
> |Method|ram.|fig.| tea.| kit.|avg.|
> |-|-|-|-|-|-|
> |Grounded SAM|14.1|16.0|16.9|16.2|15.8|
> |LangSplat|12.0|17.9|7.6|17.9|13.9|
> |SPIn-NeRF| 7.3|9.7|11.7|10.3|9.8|
> |GS-Grouping| 27.9|8.6|14.8|6.3|14.4|
> |GOI|27.1|16.5|22.9|15.7|20.5|
> |**ReferSplat**|**35.2**|**25.7**|**31.3**|**24.4**|**29.2**|
>
> [b] GOI: Find 3D Gaussians of Interest with an Optimizable Open-vocabulary Semantic-space Hyperplane, ACM MM 2024.
>
> >**Q4: Position information**
>
> **A4:** Thank you for your insightful question. Incorporating positional information into the referring feature (Eq. 7) enhances the model’s ability to understand **spatial relationships** described in the referring expressions while also enriches the referring feature with **geometric context** from the 3D scene. Instead of degrading performance, this geometric context helps ensure that segmentation masks accurately cover the complete target object, resulting in improved open-vocabulary segmentation performance.

---

> > ### Comment · Reviewer_naU6 · 2025-04-06
> >
> > The authors have addressed most of my concerns, and the additional experiments further demonstrate the effectiveness of their approach. Therefore, I will raise my score.

---

> > > ### Author Response · Authors · 2025-04-06
> > >
> > > Dear Reviewer naU6,
> > >
> > > We’re truly grateful for your support and decision to raise the score. It means a lot to us and motivates us to continue improving our work. We sincerely appreciate the time and effort you devoted to evaluating our work. We will carefully incorporate your suggestions in the final revision.
> > >
> > > Best regards,
> > > Authors of Paper #779

---

### Official Review · Reviewer_Btgt · 2025-02-21

**Overall Recommendation:** 4

**Summary:**

This paper formulates the Referring 3D Gaussian Splatting Segmentation (R3DGS) task, which focuses on segmenting 3D entities that correspond to a given referring-expression in the form of a language-based query. The R3DGS differs by the currently employed task formulation of open-vocabulary 3D segmentation by its focus on spatial relationships between scene objects, as well as distinguishing properties of objects. The paper proposes a 3D Gaussian Splatting based method, ReferSplat, which learns a referring field enabling segmentation in a multi-modal setting. To evaluate the proposed method, the paper shows results both on the R3DGS task, and the open-vocabulary 3D segmentation task. The paper extends the existing LERF dataset with annotations for referring expressions, and constructs a dataset for the R3DGS task, namely the Ref-LERF dataset, which is then used for the R3DGS evaluations.

### Update after rebuttal
The clarifications and discussions provided in the rebuttal have sufficiently addressed my concerns. I am still leaning towards acceptance, and will be keeping my original score (4: Accept).

**Claims And Evidence:**

Most claims made in the submission are supported by convincing evidence and thorough analysis. However, there are a few claims that I found not fully-founded. It might be that I interpreted certain aspects incorrectly, but I believe these statements might be a bit problematic:

1. _[L010-013] "We introduce Referring 3D Gaussian Splatting Segmentation (R3DGS), a new task that focuses on segmenting target objects in a 3D Gaussian scene based on natural language descriptions.":_ Open-vocabulary segmentation methods based on 3D Gaussian splatting representation already segment target objects based on natural language descriptions, as they take a free-form query text as input. I acknowledge that the referring expression-based segmentation is a different task as it focuses more on inter-object relations and object properties, but this statement is slightly misleading as the novel aspect of the proposed task is not segmenting objects based on "natural language descriptions", instead the referring segmentation aspect.
2. _[L034-037, Column 2] - "… identify newly described objects, even when occluded or not directly visible in the novel view…":_ The illustration and the explanation are a bit confusing as these objects are indeed visible in _some_ views. The narrative of the method being able to identify and segment objects not directly visible is not fully founded.
3. _[L074-078] - "Since the text query is only introduced during inference, the final predictions solely rely on 2D rendered features in a single-view reasoning framework, limiting the model’s ability to effectively localize objects in 3D space.":_ The limitation of these methods is indeed due to the fact that they rely on 2D rendered features instead of performing the localization directly in the 3D space. However, this is not due to "the text query only being introduced during inference" as the first part of the sentence implies.

**Essential References Not Discussed:**

I found that this submission generally discussed essential references in the context of open-vocabulary 3D Gaussian splatting segmentation. While I understand that this method has a main focus on 3DGS and not on 3D point cloud representations, given the really large body of work for 3D referring segmentation of point clouds, I found the discussion in L134-148 a bit underdeveloped. I think at least the datasets for this task (such as ScanRefer and MultiRefer) can be discussed further, to provide a better context on the relevancy of the proposed RefLERF dataset.

**Experimental Designs Or Analyses:**

As there are no available datasets for evaluating 3D referring expression segmentation task in the context of 3D Gaussian splatting, this paper proposes an extension of the LERF dataset. Experimental analysis is presented on the proposed Ref-LERF dataset. Experimental design is generally meaningful, and I found that the claims made while discussing the experimental results are reasonable.

**Methods And Evaluation Criteria:**

Proposed method is reasonable and has been designed with a meaningful thought progression, successfully targeting the 3D referring expression based segmentation task. Introducing a referring field in the representation is quite meaningful, and it addresses the limitation of existing open-vocabulary 3D Gaussian splatting methods which mainly focus on learning an implicit semantic field, often falling short on identifying inter-object spatial relations.

As there are no available datasets for evaluating 3D referring expression segmentation task in the context of 3D Gaussian splatting, this paper proposes an extension of the LERF dataset. The proposed dataset, Ref-LERF, aims to provide a reasonable evaluation benchmark for the proposed task of 3D referring expression-based segmentation for 3D Gaussian splatting. In addition to designing this dataset and performing reasonable evaluations on this dataset, the proposed method is also compared against existing benchmarks for open-vocabulary 3D segmentation for completeness. Overall, I think the evaluation is overall meaningful and through, and I have the impression that a reasonable effort was made to obtain fair comparisons.

**Other Comments Or Suggestions:**

1. _[Fig. 1 caption] "The one sanding on the..."_ should be _"standing"_
2. _[L156] "Each Gaussian g_i is parameterized its mean position..."_ should be _"by its mean position..."_
3. _[L250-251] "The proposed Position-aware Cross-Modal Interaction modules establishes"_ - should be _"module establishes"_
4. _[L040-042, Column 2] - "A straightforward baseline for R3DGS is to adapt existing open-vocabulary 3D scene understanding methods by replacing the open-vocabulary class names with natural language expressions.":_ I agree with this statement in the sense that it accurately identifies how a straightforward baseline can be formed from open-vocabulary 3D scene understanding methods. However, the phrasing "open-vocabulary class names" is not very accurate.
5. The impact statement required by ICML is missing in this submission.

**Other Strengths And Weaknesses:**

Strengths:
- The paper addresses an important topic (3D referring segmentation) that was not explored in the context of 3DGS representations to the best of my knowledge.
- The paper is generally written with clarity.
- Qualitative examples are interesting, I appreciated that the examples in Figure 5 generally feature expressions where the target object's semantics are not explicitly stated, showing the strengths of the method for identifying objects mostly based on object-relations.

Weaknesses:
- If I understand correctly, the method requires the generation of a set of referring expression sentences first, which are then used during training. However, it is not clear to me how or when these sentences are generated. How are these sentences generated?
- I am unclear about how well the method can generalize to new referring expressions if it is trained with a set of pre-written expressions. I understand that the contrastive module is introduced for this purpose, but I do not understand how it is possible to generalize if the model never sees certain types of object relationship descriptors.
- L196-200 state that the overall similarity is measured by summing per-word similarities. But how are tokens representing negative relations accounted for? For instance if the query is "the object far away from the window", I still suspect that the model will identify an object right by the window. How are such cases handled in the model implicitly?
- I understand that based on the training referring expression sentences, pseudo-GT 2D masks are generated using Grounded-SAM. This means that the method indeed relies on the use of 2D masks. However there are some statements claiming that the proposed method circumvents using masks. This is not fully founded, and a bit confusing.
- For extracting word and sentence features, the method employs BERT embeddings (L314, right column) despite using CLIP distillation at another part of the method (L196, right column). I was unable to see any discussion or an ablation regarding the choice of BERT instead of CLIP to extract features from the text input.

**Questions For Authors:**

Please see the questions listed in Strengths and Weaknesses.

**Relation To Broader Scientific Literature:**

Referring expression segmentation is a crucial task for many robotics applications. While there is a large body of work towards this goal in the 3D scene understanding domain that primarily focus on 3D point cloud representations, I am not aware of any works addressing the same for higher-fidelity representations such as 3D Gaussian splatting-based methods. There is another line of work for addressing natural-language based segmentation of 3D Gaussian splatting representations, particularly in the context of open-vocabulary segmentation. However, the design of such methods as well as the evaluation methodology is generally more fixated on identifying objects based on their semantics. Often, those methods are evaluated using a set of text queries describing the target object- however, to the best of my knowledge none of these methods systematically evaluate for how well the method can identify objects based on their relations to other scene objects. By extending the existing LERF dataset, this submission takes a step towards a relatively less explored aspect of language-guided segmentation of 3D Gaussian splatting representations.

**Theoretical Claims:**

I did not identify any theoretical claims with proofs provided in the submission.

---

> ### Author Rebuttal · Authors · 2025-04-01
>
> We sincerely appreciate your positive feedback on our work: meaningful, reasonable, and thorough.
>
> >**Q1: Claim 1**
>
> **A1:** In established 2D/3D referring expression segmentation (RES) tasks, referring segmentation involves segmenting target objects based on free-form natural language expressions that often include spatial relationships or object attributes. In contrast, open-vocabulary segmentation typically focuses on identifying all objects of a given category, usually specified by a single **category name**. As shown in Fig. 4 of our paper, the average sentence length in the LERF-OVS dataset is approximately **1.5 words**, with the vast majority of queries being category names (e.g., “cup”) and very few containing descriptive or relational information (e.g., “tea in a glass”). This highlights a clear difference in the nature of the language inputs between the two tasks.
> To avoid confusion and better reflect the novelty of our task, we have revised the sentence to:
> “*a new task that focuses on segmenting target objects in a 3D Gaussian scene based on natural language descriptions that often contain spatial relationships or object attributes.*”
>
> >**Q2: Claim 2**
>
> **A2:** While the elephant is clearly visible in some training views, it is **not directly visible** from the **novel camera viewpoint** during inference in Fig. 1. We highlight this to emphasize a key difference from 2D RES, which relies solely on single-view information and struggles to handle such invisible scenarios. In contrast, the proposed ReferSplat leverages multi-view information to construct a holistic 3D scene representation, enabling robust reasoning even when target objects are occluded or not directly visible. For further clarification, please refer to L128–137 (second column). We will revise the corresponding description to make this point clearer.
>
> >**Q3: Claim 3**
>
> **A3:** We agree with your comment and have revised the sentence for better clarity. The updated version is:
> *“Existing methods rely on matching the text query with 2D rendered features instead of performing localization directly in 3D space, which limits their performance in complex scenarios.”*
>
> >**Q4: Related work**
>
> **A4:** Thank you for the suggestion. We will expand the related work to provide more discussion on datasets like ScanRefer and MultiRefer.
>
> >**Q5: Referring expression sentences generation**
>
> **A5:** The referring expressions are manually annotated by human annotators prior to training and are included in our Ref-LERF dataset.
>
> >**Q6: Generalize to new referring expressions**
>
> **A6:** For generalization to new expression, our method uses BERT embeddings for their strong language understanding and generalization, gained via pre-training on diverse text. By aligning 3D Gaussian features with BERT representations via Gaussian-Text modeling, the model can interpret and generalize to unseen object relationship descriptors. We will clarify this in the revision. For generalization to new scenes and new datasets, please refer to our responses to Reviewer 2Vdk Q4 and Reviewer t3nW Q5, respectively.
>
> >**Q7: Overall similarity**
>
> **A7:** Thank you for the insightful question. Our method aggregates per-word similarities, allowing the model to account for the influence of individual terms, including spatial relations like “far away.” Visualizations show that such words activate relevant regions, suggesting the model captures their importance in segmentation decisions. Additionally, we incorporate Gaussian-Text Contrastive Learning on global sentence-level features to enhance **holistic comprehension of the sentence context**, including complex spatial relationships. This combination of **local (word-level) and global (sentence-level) cues** enables the model to better interpret both positive and negative spatial terms. We will clarify this aspect in the revision.
>
> >**Q8: Mask usage**
>
> **A8:** Our method does employ pseudo masks generated by Grounded-SAM during training. The intended point is to emphasize that our method eliminates the need for manually annotated ground-truth masks, which are often costly and impractical. We will revise any ambiguous statements in the paper to more accurately reflect this and avoid confusion.
>
> >**Q9: CLIP result**
>
> **A9:** As suggested, we conduct experiments comparing BERT and CLIP embeddings for language features in R3DGS. Results show that BERT consistently outperforms CLIP. This is likely because CLIP focuses more on noun categories, while referring expressions often involve spatial and attribute-based descriptions. We will include more discussion and results in the revision.
> |Method|ram.|fig.| tea.| kit.|avg.|
> |-|-|-|-|-|-|
> |BERT|35.2| 25.7 |31.3 |24.4 |29.2|
> |CLIP|23.5|23.2|26.2 | 21.0|23.5|
>
> >**Q10: Typos**
>
> **A10:** Thank you! We will carefully proofread and correct any typos or ambiguous phrasing in the revision.
>
> >**Q11: Impact statement**
>
> **A11:** We will add impact statement in the revision.

---

> > ### Comment · Reviewer_Btgt · 2025-04-03
> >
> > I thank the authors for the rebuttal. The clarifications and discussions provided in the rebuttal have sufficiently addressed my concerns. I am still leaning towards acceptance, and will be keeping my original score (4: Accept).

---

> > > ### Author Response · Authors · 2025-04-03
> > >
> > > Dear Reviewer Btgt,
> > >
> > > Thank you for your thoughtful review and great support. We are glad that our rebuttal has addressed your concerns, and we sincerely appreciate the time and effort you devoted to evaluating our work. We will carefully incorporate your suggestions in the final revision.
> > >
> > > Best regards,
> > > Authors of Paper #779

---

### Official Review · Reviewer_2Vdk · 2025-03-02

**Overall Recommendation:** 3

**Summary:**

The paper introduces ReferSplat, a framework for Referring 3D Gaussian Splatting Segmentation (R3DGS), aiming to segment 3D objects based on **natural language descriptions**, even when occluded or not directly visible. Key contributions include:

1. **R3DGS Task**: A new task requiring 3D multi-modal understanding and spatial reasoning.
2. **Ref-LERF Dataset**: A dataset with 295 language expressions emphasizing spatial relationships and object attributes.
3. **ReferSplat Framework**: Combines 3D Gaussian Referring Fields, Position-aware Cross-Modal Interaction (PCMI), and Gaussian-Text Contrastive Learning (GTCL) to align language and 3D spatial features.
4. **State-of-the-Art Results**: Outperforms existing methods on R3DGS and 3D open-vocabulary segmentation benchmarks.

**Claims And Evidence:**

- ReferSplat’s superiority over baselines (LangSplat, SPIn-NeRF) is validated through quantitative results (Tables 5-7).
- Ablation studies (Tables 1-4) confirm the effectiveness of PCMI and GTCL.
- Pseudo mask generation via confidence-weighted IoU improves mask quality (Table 3).

**Essential References Not Discussed:**

No.

**Experimental Designs Or Analyses:**

- **Strengths**:
    - Comprehensive ablation studies validate each component.
    - Failure case analysis (Appendix E) highlights practical challenges.
- **Weaknesses**:
    - Dataset splits (train/test) and generalization to unseen scenes are unclear.
    - without training & rendering time comparsion
    - without storage comparsion

**Methods And Evaluation Criteria:**

- **Strengths**:
    - The integration of language features into 3D Gaussians via referring fields is novel and well-motivated.
    - PCMI and GTCL address spatial reasoning and semantic disambiguation effectively.
    - This paper is well-written.
- **Weaknesses**:
    - Ref-LERF’s limited scale (only LERF-OVS dataset, including 4 scenes and 59 objects) and lack of diversity comparison to existing datasets (e.g., Scannet/Scannet++) may limit generalizability.
    - Evaluation focuses on mIoU but omits metrics like precision/recall for occlusion cases.

**Other Comments Or Suggestions:**

No

**Other Strengths And Weaknesses:**

No

**Questions For Authors:**

No

**Relation To Broader Scientific Literature:**

I'm not familiar with Segmentation + 3DGS, so I have no idea whether Refersplat is related to previous papers.

**Theoretical Claims:**

No

---

> ### Author Rebuttal · Authors · 2025-04-01
>
> We sincerely appreciate your positive feedback on our work: novel, well-motivated, effective PCMI and GTCL design, and well-written.
>
> >**Q1: Diversity comparison to existing datasets**
>
> **A1:** The comparision to ScanRefer and Multi3DRefer is shown in the table below. Due to the limited availability of large-scale 3DGS datasets, our Ref-LERF is built upon the widely used LERF dataset, extended with referring sentence annotations to support the R3DGS task. While the number of scenes is limited, Tab. 5 of the main paper shows that our method achieves strong performance, demonstrating its effectiveness. Furthermore, as discussed in our response to Q4, our method also generalizes well to larger and more diverse datasets, e.g., ScanNet, under the generalized training setting.
> |Dataset|Year|Pub.|#Object|#Expression|#scenes|data format|
> |-|-|-|-|-|-|-|
> |ScanRefer|2020|ECCV|11,046|51,583|800|3D scan|
> |Multi3DRefer|2023|ICCV|11,609 |61,926|800|3D scan|
> |Ref-LERF|2025|-|59|295|4|Multi-view Images|
>
> >**Q2: Evaluation metric**
>
> **A2:** To provide a more comprehensive evaluation, we report mAcc@0.25, which measures the percentage of predictions with IoU > 0.25 and is commonly used in 3D point cloud referring segmentation tasks. This metric reflects performance in practical scenarios, including occlusions. As shown in the table below, ReferSplat achieves a mean mAcc. of 38.4 on Ref-LERF, significantly outperforming LangSplat and demonstrating our method’s robustness. We will include this metric and the results in the revision.
> |Method|ram.|ram.|fig.|fig.|tea.|tea.|kit.|kit.|avg.|avg.
> |-|-|-|-|-|-|-|-|-|-|-|
> |Metric|mIoU|mAcc.|mIoU|mAcc.|mIoU|mAcc.|mIoU|mAcc.|mIoU|mAcc.|
> |LangSplat |12.0|18.4|17.9|25.4| 7.6|10.2| 17.9|27.5| 13.9|20.4|
> |GS-Grouping |27.9|30.3 |8.6|11.1| 14.8|16.9| 6.3|13.8| 14.4|18.0|
> |**ReferSplat**|**35.2**|**50.0**|**25.7**|**31.7**| **31.3**|**33.9**| **24.4**|**37.9**|**29.2**|**38.4**|
>
> >**Q3: Dataset splits**
>
> **A3:** The dataset comprises 772 training images and 22 testing images, with 236 language descriptions used for training and 59 for testing, totaling 295 descriptions. We will add the detailed dataset split information in the revision.
>
> >**Q4: Generalization to unseen scenes**
>
> **A4:** The experiments in our main paper follow a per-scene optimization setup, which naturally limits direct generalization to unseen scenes. To address this, we explore a generalized training paradigm, as shown in **Tab. 11 of the Appendix**. In this setting, the referring feature is no longer per-scene initialized; instead, it is predicted from other Gaussian attributes such as color, opacity, and position, following a feed-forward paradigm. This design enables generalization beyond per-scene optimization. To further validate the generalization capability of our method, we conduct additional experiments on the larger-scale and more diverse ScanNet dataset. We select 30 scenes from the official training split for joint training and 5 scenes from the validation split for evaluation. Language expressions are sourced from ScanRefer. As shown in the table below, our method achieves strong performance across diverse and unseen environments, highlighting its robustness and generalization. We expect that extending our framework with scalable architectures (e.g., MVSplat [a]) will lead to further performance gains and improved applicability in real-world scenarios.
> |Method|scene0011|scene0015|scene0019| scene0025|scene0030|mIoU|
> |-|-|-|-|-|-|-|
> |ReferSplat|15.9|19.7|27.8|18.3|21.4|20.6
>
> [a] MVSplat: Efficient 3D Gaussian Splatting from Sparse Multi-View Images, ECCV 2024.
>
> >**Q5&Q6: Computational costs**
>
> **A5:** We conduct experiments on the ramen scene from the Ref-LERF dataset using the same NVIDIA A6000 GPU to compare the computational cost of our ReferSplat against SOTA methods. Results show that ReferSplat achieves significantly lower computational complexity and faster inference speed than LangSplat. While GS-Grouping excels in storage and FPS, ReferSplat outperforms all methods in segmentation performance. ReferSplat also has the shortest training time, thanks to a lightweight preprocessing pipeline that avoids costly operations like language feature compression (LangSplat) or mask association with video tracking methods (GS-Grouping). These results demonstrate that ReferSplat’s compact, efficient design is well-suited for real-world and large-scale 3D applications. We will include these comparisons in the revision.
> Method|Training↓|FPS↑|Storage↓|mIoU↑|
> -|-|-|-|-|
> LangSplat|176min|12.4|46MB|13.9|
> GS-Grouping|66min|**54.2**|**2.3MB**|14.4|
> **ReferSplat**|**58min**|26.8|3.3MB|**29.2**|

---

> > ### Comment · Reviewer_2Vdk · 2025-04-06
> >
> > I have reviewed all the rebuttal comments, and my concerns have been satisfactorily addressed. I have no further questions at this time.

---

> > > ### Author Response · Authors · 2025-04-06
> > >
> > > Dear Reviewer 2Vdk,
> > >
> > > Thank you sincerely for your time and thoughtful review. We’re glad to hear that our rebuttal has satisfactorily addressed your concerns.
> > > If you find our clarifications reasonable, we would greatly appreciate your consideration in updating the score to Accept (4). Your support would mean a lot to us and would strongly encourage our continued work in this direction.
> > >
> > > Best regards,
> > > Authors of Paper #779

---

### Official Review · Reviewer_t3nW · 2025-03-16

**Overall Recommendation:** 3

**Summary:**

- This paper introduces Referring 3D Gaussian Splatting Segmentation (R3DGS), a task aimed at segmenting target objects in a 3D Gaussian scene based on natural language descriptions.
- The proposed method addresses key challenges, including identifying occluded objects in novel views.
- The authors present Ref-LERF, for the proposed task.
- The framework integrates 3D Gaussian referring fields, a position-aware cross-modal interaction module, and Gaussian-Text Contrastive Learning to improve spatial reasoning and enhance fine-grained understanding of natural language descriptions.


## update after rebuttal
I maintain my original rating.

**Claims And Evidence:**

The authors extend the existing LeRF dataset by incorporating expressive annotations, providing five descriptions of varying token lengths for each object. Figure 4(b) showcases the dataset’s increased complexity.

**Essential References Not Discussed:**

All the related references are cited in the manuscript.

**Experimental Designs Or Analyses:**

Yes, ablation studies in Section 4.3 discuss different design choices.

**Methods And Evaluation Criteria:**

Yes, the proposed baselines and benchmark datasets are logical.

**Other Comments Or Suggestions:**

-

**Other Strengths And Weaknesses:**

**Strengths**
- **[S1] Dataset Contribution**: The creation of the Ref-LERF dataset provides valuable resources for future research for the proposed task.

- **[S2] Outperforms SOTA methods**: The proposed model achieves state-of-the-art performance on the newly introduced R3DGS task and existing 3D open-vocabulary segmentation benchmarks.

- **[S3]** Unlike retrieval-based matching between rendered semantic features and text embeddings, the relationship is directly modelled in the proposed method.

- **[S4] Exhaustive ablations**: The authors provide thorough details regarding design choices, including the number of input views, the dimension of referring features and selection strategies for positive referring features.


**Weaknesses**

- **[W1] Detailed Evidence**: The paper lacks detailed evidence on the model's performance when handling highly ambiguous or incomplete language queries, which may limit its practicality in real-world applications. While Gaussian-Text Contrastive Learning is introduced to address ambiguities, it could still lead to confusion in referring segmentation for different objects. Additional ablations, videos, or novel view evaluations would strengthen the paper's claims and provide clearer validation of its effectiveness.

- **[W2] Training time**: The paper does not thoroughly address the method's training and inference time. The method's complexity, particularly the position-aware cross-modal interaction module, may result in high computational costs, potentially limiting its feasibility for large-scale or real-time 3D environments.

- **[W3] Difficult with sudden viewpoint changes**: The authors state that the model struggles with significant viewpoint changes and perspective shifts. However, how do other baselines perform under these conditions? Is this limitation specific to the proposed method, or is it a broader challenge in the field?

**Questions For Authors:**

- Fig 1. The elephant is clearly visible in the scene, so why do the authors claim that it is not?

- How would the model perform on datasets like MessyRooms, where some scenes contain up to 1,000 objects? Given that manual annotation and descriptions are impractical in such cases, how can this challenge be addressed?

- The approach relies heavily on pseudo-ground truth masks. How does the proposed method ensure that errors in these masks do not negatively impact the model's performance?

- Could the authors provide more qualitative results or supporting videos showcasing the model's performance from different novel views? This would further strengthen the paper's claims and demonstrate the method's effectiveness.

**Relation To Broader Scientific Literature:**

This paper advances the challenge of 3D segmentation by integrating natural language descriptions for object identification, even in cases where objects are occluded or invisible from a single view. To support this task, the authors introduce Ref-LERF, a novel dataset specifically designed for language-guided 3D segmentation. This contribution is valuable to the scientific community, enabling more robust and interpretable scene understanding.

**Theoretical Claims:**

There is no proof in the manuscript.

---

> ### Author Rebuttal · Authors · 2025-04-01
>
> We sincerely appreciate your positive feedback on our work: dataset contribution, SOTA result, relationship modeling, and exhaustive ablations.
>
> >**Q1&Q7: Detailed evidence like video**
>
> **A1:** We have provided additional qualitative video results at **[ReferSplat.mp4](https://anonymous.4open.science/api/repo/ReferSplat-779/file/ReferSplat.mp4)**, which include:
> * Cases involving ambiguous language queries
> * Cases with incomplete language input
> * Performance under different novel views
> * Performance on significant viewpoint and perspective shift
>
> These videos demonstrate our method’s ability to handle diverse and challenging referring expressions, showcasing its robustness to ambiguity and incomplete descriptions. The qualitative results further validate the effectiveness of our method and its potential for real-world applications.
>
> >**Q2: Computational costs**
>
> **A2:** We conduct experiments on the ramen scene from the Ref-LERF dataset using the same NVIDIA A6000 GPU to compare the computational cost of our ReferSplat against SOTA methods. Results show that ReferSplat achieves significantly lower computational complexity and faster inference speed than LangSplat. While GS-Grouping excels in storage and FPS, ReferSplat outperforms all methods in segmentation performance. ReferSplat also has the shortest training time, thanks to a lightweight preprocessing pipeline that avoids costly operations like language feature compression (LangSplat) or mask association with video tracking methods (GS-Grouping). These results demonstrate that ReferSplat’s compact, efficient design is well-suited for real-world and large-scale 3D applications. We will include these comparisons in the revision.
> Method|Training↓ |FPS↑|Storage↓|mIoU↑|
> -|-|-|-|-|
> LangSplat|176min|12.4|46MB|13.9|
> GS-Grouping|66min|**54.2**|**2.3MB**|14.4|
> **ReferSplat**|**58min**|26.8|3.3MB|**29.2**|
>
> >**Q3: Difficult with sudden viewpoint changes**
>
> **A3:** This is a broader challenge in the field, not specific to our method. Baselines like LangSplat and GS-Grouping also show performance degradation under significant viewpoint changes and perspective shifts, as shown in the **[ReferSplat.mp4](https://anonymous.4open.science/api/repo/ReferSplat-779/file/ReferSplat.mp4)**. Future work will focus on developing robust multimodal representations and improving global scene understanding to enhance robustness under extreme viewpoint variations.
>
> >**Q4: Elephant visible?**
>
> **A4:** While the elephant is clearly visible in some training views, it is **not directly visible** from the **novel camera viewpoint** during inference in Fig. 1. We highlight this to emphasize a key difference from 2D referring expression segmentation (RES), which relies solely on single-view information and struggles to handle such invisible scenarios. In contrast, the proposed ReferSplat leverages multi-view information to construct a holistic 3D scene representation, enabling robust reasoning even when target objects are occluded or not directly visible. For further clarification, please refer to L128–137 (second column). We will revise the corresponding description to make this point clearer.
>
> >**Q5: Messy Rooms dataset evaluation**
>
> **A5:** Drawing from experience in 2D/3D RES tasks, models trained on sufficiently diverse datasets can generalize well and exhibit strong zero-shot capabilities, as exemplified by models like Grounded-SAM. As shown in Appendix Tab. 11, our model shows promising generalization results under the generalized training setting. In particular, it generalizes well to unseen scenes on the ScanNet dataset, which contains numerous objects and diverse scene layouts (see Reviewer 2Vdk Q4). Building upon this foundation, joint training on large-scale, diverse datasets such as ScanRefer, Ref-LERF, and MultiRefer presents a viable path toward adapting our model to more complex environments like Messy Rooms, where manual annotation and detailed descriptions are impractical. The above analysis highlights the scalability and cross-dataset generalization potential of our approach in real-world applications.
>
> >**Q6: Pseudo mask error**
>
> **A6:** To reduce the impact of errors from pseudo masks, we employ a two-stage optimization strategy (as described in L310-316), iteratively refining mask predictions during training. As shown in Tab. 1 of the main paper, this two-stage strategy outperforms the one-stage pipeline, demonstrating its effectiveness in mitigating the impact of noisy pseudo masks and ensuring robust overall performance. It is worth noting that the default results are based on the one-stage pipeline for a fair comparison with previous methods.

---

> > ### Comment · Reviewer_t3nW · 2025-04-03
> >
> > I have reviewed all the rebuttal comments, and my queries have been satisfactorily addressed. I have no further questions.

---

> > > ### Author Response · Authors · 2025-04-03
> > >
> > > Dear Reviewer t3nW,
> > >
> > > Thank you for your thoughtful review and positive acknowledgment. We sincerely appreciate your constructive feedback, which has helped us improve the clarity and quality of our paper. We will carefully incorporate your suggestions in the final revision.
> > >
> > > Best regards,
> > > Authors of Paper #779

---

### Decision · Program_Chairs · 2025-05-01

**Decision:**

Accept (oral)

**Comment:**

This work initially received unanimously positive reviews in major aspects. It aims for an interesting and well-motivated 3D task while contributing a newly introduced dataset, outperforming previous methods, with more directly applied text embeddings for their referring segmentation in the 3DGS task, with extensive ablation studies.

Some raised concerns include detailed evidence, time complexity, generalization towards novel views, and natural language expressions. The authors faithfully provided the requested experiments and reasonable explanations, which led to satisfactory responses from all reviewers. AC thinks this quality paper got quality reviews and exceptionally constructive rebuttals to help improve the manuscript.

AC also leave minor comments on visualization and table style.
* In Figure 5, why is the Grounded-SAM result for "ramen" horizontally flipped?
* Tables 1-7 have narrow space margins between captions and tables.
* Missing running titles on the top of pages.

Overall, AC appreciates this well-written and hardworking paper and happily recommends this paper for acceptance to our program.